# In-Vitro Selection of Ceftazidime/Avibactam Resistance in OXA-48-Like-Expressing *Klebsiella pneumoniae*: In-Vitro and In-Vivo Fitness, Genetic Basis and Activities of β-Lactam Plus Novel β-Lactamase Inhibitor or β-Lactam Enhancer Combinations

**DOI:** 10.3390/antibiotics10111318

**Published:** 2021-10-29

**Authors:** Snehal Palwe, Yamuna Devi Bakthavatchalam, Kshama Khobragadea, Arun S. Kharat, Kamini Walia, Balaji Veeraraghavan

**Affiliations:** 1Department of Environmental Science, S. B. College of Science, Aurangabad 431001, India; snehalpalwe7@gmail.com (S.P.); kskhobragade@sbscience.org (K.K.); 2Department of Clinical Microbiology, Christian Medical College, Vellore 632004, India; cilviamicrovin@gmail.com; 3Laboratory of Applied Microbiology, Jawaharlal Nehru University, New Delhi 110067, India; arunkharat2007@gmail.com; 4Division of Epidemiology and Communicable Diseases, Indian Council of Medical Research, New Delhi 110029, India; waliakamini@gmail.com

**Keywords:** ceftazidime/avibactam, β-lactams, β-lactamase inhibitor, multi-drug resistance

## Abstract

Ceftazidime/avibactam uniquely demonstrates activity against both KPC and OXA-48-like carbapenemase-expressing Enterobacterales. Clinical resistance to ceftazidime/avibactam in KPC-producers was foreseen in in-vitro resistance studies. Herein, we assessed the resistance selection propensity of ceftazidime/avibactam in *K. pneumoniae* expressing OXA-48-like β-lactamases (*n* = 10), employing serial transfer approach. Ceftazidime/avibactam MICs (0.25–4 mg/L) increased to 16–256 mg/L after 15 daily-sequential transfers. The whole genome sequence analysis of terminal mutants showed modifications in proteins linked to efflux (AcrB/AcrD/EmrA/Mdt), outer membrane permeability (OmpK36) and/or stress response pathways (CpxA/EnvZ/RpoE). In-vitro growth properties of all the ceftazidime/avibactam-selected mutants were comparable to their respective parents and they retained the ability to cause pulmonary infection in neutropenic mice. Against these mutants, we explored the activities of various combinations of β-lactams (ceftazidime or cefepime) with structurally diverse β-lactamase inhibitors or a β-lactam enhancer, zidebactam. Zidebactam, in combination with either cefepime or ceftazidime, overcame ceftazidime/avibactam resistance (MIC range 0.5–8 mg/L), while cefepime/avibactam was the second best (MIC: 0.5–16 mg/L) in yielding lower MICs. The present work revealed the possibility of ceftazidime/avibactam resistance in OXA-48-like *K. pneumoniae* through mutations in proteins involved in efflux and/or porins without concomitant fitness cost mandating astute monitoring of ceftazidime/avibactam resistance among OXA-48 genotypes.

## 1. Introduction

The carbapenem non-susceptibility in Enterobacterales is mainly imparted by the ‘triad’ of carbapenemases, viz., *Klebsiella pneumoniae* carbapenemases (KPCs), metallo-β-lactamases (MBLs) and OXA-48-like. Lately, OXA-48-like carbapenemases have captured the attention due to their wider geographic dissemination leading to their ascendency in several parts of the globe, including the Middle East, North Africa, some European countries and the Indian sub-continent [1]. Among the reported multiple plasmid-encoded blaOXA-48-like variants, the prototype blaOXA-48-like, its four-amino-acid variant-blaOXA-181 and the five-amino-acid variant-blaOXA-232 are predominant. Compared to the other two carbapenemases in the triad, OXA-48-like β-lactamases possess a weaker carbapenemase activity; however, when they co-exist with impermeability, extended-spectrum β-lactamases (ESBLs) and/or AmpC, a clinically-relevant resistance to carbapenems is observed [1,2,3,4]. In particular, such a multiplicity of resistance mechanisms, usually prevalent in *K. pneumoniae* harboring OXA-48-like, presents a therapeutic challenge to diverse antibiotic classes [1,5,6].

Disappointingly, recently approved carbapenem based combinations, imipenem/relebactam and meropenem/vaborbactam, do not cover OXA-48-like-producing *K. pneumoniae* due to the inability of respective β-lactamase inhibitors to inhibit these enzymes [7,8]. Though cephalosporins are stable to OXA-48-like, the coverage of this resistotype by cephalosporin-based combinations is contingent to the β-lactam’s vulnerability towards co-expressed β-lactamases, which in turn is linked with partner β-lactamase inhibitor’s ability to restore the activity of partner β-lactam to a therapeutically meaningful level. This resonates in the activity of ceftazidime/avibactam against OXA-48-like-expressing *K. pneumoniae* but not for ceftolozane/tazobactam [8,9]. Thus, currently, ceftazidime/avibactam is the only approved β-lactam (BL)/β-lactamase inhibitor (BLI) [BL/BLI] combination with coverage of OXA-48-like-expressing organisms. Recently, registered cefiderocol has been reported to be active against isolates expressing OXA-48-like [10]; however, in the absence of definitive diagnosis of the ‘type’ of carbapenem-resistant Enterobacterales (CRE) in most clinical settings, treatment decisions need to factor a wide scatter in cefiderocol MIC_90_ for isolates with New Delhi metallo-β-lactamases (NDM) often exceeding the susceptibility breakpoint [10,11]. Among the pipeline agents, cefepime/taniborbactam, cefepime/enmetazobactam and cefepime/zidebactam have been shown to possess in-vitro and in-vivo activity against pathogens expressing this resistance mechanism [12,13,14].

In regions with predominant prevalence of KPC, such as the USA, the clinical use of ceftazidime/avibactam is mainly directed towards confirmed or suspected cases of KPC infection. Unfortunately, within a few years of its usage, reports of on-therapy resistance began to multiply in KPC-expressing *K. pneumoniae* which was associated with mutations in KPC leading to enhanced hydrolytic activity for ceftazidime rendering avibactam’s inhibitory activity inadequate [15,16,17,18,19]. The early signals of propensity of resistance development with ceftazidime/avibactam against KPC were detected in the in-vitro studies that showed relative ease of acquisition of mutational resistance in single and multiple drug-passage studies [20]. However, in-vitro studies assessing the propensity of resistance development to ceftazidime/avibactam against OXA-48-like resistotype are extremely limited, specifically those employing ‘real-world’ clinical isolates of *K. pneumoniae*.

To the best of our knowledge, this is the first ever study assessing the phenotypic and genotypic aspects of in-vitro resistance development to ceftazidime/avibactam against clinical isolates of *K. pneumoniae* harboring blaOXA-181 or blaOXA-232 and their in-vivo relevance. Additionally, to explore the interplay between β-lactams and β-lactamase inhibitors in tackling potential acquisition of ceftazidime/avibactam resistance in OXA-48-like pathogens, we evaluated the activities of structurally and mechanistically diverse BLIs as well as β-lactam enhancer, zidebactam, when partnered with cefepime or ceftazidime.

## 2. Results

### 2.1. Serial Transfer Study

The OXA-48-like-expressing *K. pneumoniae* isolates utilized in this study were susceptible to ceftazidime/avibactam as shown in Table 1 (MICs 0.25 to 4 mg/L; susceptibility breakpoint 8 mg/L). Figure 1 shows the day-wise changes in the MICs of ceftazidime/avibactam during the 15 sequential passages. The rate of MIC increase was variable in the range of 3 to 15 passages to raise the MICs above 8 mg/L; generally higher passage numbers were required for strains with lower baseline MICs. The MICs after the last passage ranged from 16 to 256 mg/L, all resistant to ceftazidime/avibactam. The resistance to ceftazidime/avibactam in the terminal mutants was stable after 15 drug-free passages (MICs 16—>128 mg/L). Furthermore, all the terminal mutants grew well, comparable to their respective parent isolates in M9 minimal medium and cation-adjusted Mueller–Hinton broth (CAMHB), suggesting lack of fitness cost (Appendix A).

### 2.2. Whole Genome Sequencing

Table 2 shows the mutations identified through whole genome sequencing analysis of ceftazidime/avibactam-selected terminal mutants when compared to the respective parent isolates. Broadly, changes were observed in proteins linked to efflux (AcrB/AcrD/EmrA/Mdt), outer membrane permeability (OmpK36) and/or stress response pathways (CpxA/EnvZ/RpoE). In one of the mutants (derived from *K. pneumoniae* S465), point mutation in AmpC β-lactamase was detected that might have increased its hydrolytic activity towards ceftazidime leading to elevated ceftazidime/avibactam MICs. Furthermore, two blaOXA-181-producing isolates (*K. pneumoniae* AI1547 and *K. pneumoniae* S471) selected point mutations leading to conversion of blaOXA-181 to blaOXA-232 in addition to the mutations in proteins related to efflux and outer membrane permeability, respectively. Current and previously reported studies provide support to the fact that ceftazidime/avibactam retains activity against all variants of blaOXA-48-like such as blaOXA-232, blaOXA-162 etc. expressing *K. pneumoniae* [21] and therefore, the observed change in OXA-enzyme type (from 181 to 232) may not be a standalone cause for increase in ceftazidime/avibactam MICs. Lastly, ceftazidime/avibactam selected AI1235 mutant showed mutation in MrdA (penicillin binding protein [PBP] 2) as previously reported [22]. This suggests PBP2 sensitization, most probably, by avibactam, which is known to have a weak affinity towards PBP2; however, the role of this mutation in elevating ceftazidime/avibactam MIC for this isolate is unlikely since as compared to *E. coli*, the intrinsic potency of avibactam against *K. pneumoniae* is of a lower order.

### 2.3. In-Vivo Infectivity and Resistance

In order to assess the in-vivo relevance of the raised ceftazidime/avibactam MICs, the magnitude of protective doses (measured as efficacy dose 50/90, ED50/90) of ceftazidime/avibactam were compared for parent vs. mutant (for two parent-mutant pairs) in a mice peritonitis model. Untreated control animals succumbed within 24 h of infection, suggesting optimal infectivity of both parent and mutants. Against two parent strains, ceftazidime in combination with avibactam (4:1 dose ratio) showed ED50 of 6 to 14.9 mg/kg which increased to 34.4 to 49.5 mg/kg for their respective mutants, thus reflecting the ceftazidime/avibactam resistance acquired by the terminal mutants (Figure 2). Similar, increase in ED90 of ceftazidime/avibactam was also observed in mutants (80.7–102.5 mg/kg) compared to parent strains (20.9–34.6 mg/kg). All the animals in meropenem treatment groups of both parent and mutants showed mortality within 24 h of infection suggesting optimal carbapenemase expression.

### 2.4. Susceptibility Profile of Serial Transfer Mutants to Combinations of Ceftazidime or Cefepime with β-Lactamase Inhibitors/β-Lactam Enhancer

To assess the interplay between β-lactam and β-lactamase inhibitors/β-lactam enhancer in tackling the acquired resistance to ceftazidime/avibactam in OXA-48-like expressing *K. pneumoniae*, we determined the MICs of three cefepime based combinations under clinical development, two exploratory cefepime/BLI combinations, as well as various exploratory combinations of ceftazidime with BLI or β-lactam enhancer (Table 3). Carbapenems were included in the MIC studies to determine the collateral effect of acquired resistance on their activity, if any.

The MICs of carbapenems for the mutants were generally comparable to respective parents; limited to 1 to 2 dilution-fold change in MICs. The lone exception was the mutant of parent AI1646 for which MICs of meropenem and imipenem substantially dropped to susceptibility range, a phenomenon commonly observed with KPC harboring *K. pneumoniae* that acquired resistance to ceftazidime/avibactam [23]. Overall for all parent strains, avibactam lowered the MICs of imipenem to near baseline, while for mutants, the combination MICs were within 1 to 2 dilution folds compared to parent isolates (highest MIC being 1 mg/L except for the isolate CMC307).

There was a near complete cross-resistance carried to all ceftazidime based exploratory combinations except for ceftazidime/zidebactam (MIC range: 0.5–2 mg/L for parents and 0.5–8 mg/L for mutants), wherein zidebactam is a β-lactam enhancer. In case of cefepime based combinations, overall MICs were lower compared to the respective ceftazidime-based combination and among all, cefepime/zidebactam (MIC range: 0.5–2 mg/L for parents and 0.5–8 mg/L for mutants) and cefepime/avibactam (MIC range: 0.25–2 mg/L for parents and 0.5–16 mg/L for mutants) were the most potent.

From the therapeutic perspectives, none of the ceftazidime-BLI combinations yielded MICs of ≤8 mg/L, a hypothetical cut-off applied based on ceftazidime/avibactam breakpoint, and in contrast, majority of mutants showed cefepime/avibactam MICs of ≤8 mg/L, employing cefepime’s susceptibility dose-dependent breakpoint. In combination with zidebactam, both cefepime and ceftazidime yielded MICs ≤ 8 mg/L against all the ceftazidime/avibactam-resistant mutants.

## 3. Discussion

The launch of ceftazidime/avibactam in 2015 aptly generated a high anticipation among the clinicians owing to its activity against challenging resistance mechanisms expressed in Enterobacterales; AmpC, OXA-48-like and KPC β-lactamases, which are not amenable to then available β-lactam/β-lactamase-inhibitor combinations. However, within 5 years of ceftazidime/avibactam’s clinical use, several disturbing reports, revealing its vulnerability to on-therapy resistance selection in KPC-expressing pathogens, began appearing [16,24,25]. Corroborating this trend, in-vitro resistance selection studies published earlier have pointed towards the risk of clinical resistance to ceftazidime/avibactam, thus signifying the relevance of such in-vitro resistance selection studies [20,22].

In view of the therapeutic value of ceftazidime/avibactam for the infections caused by OXA-48-like producers, we sought to decipher the propensity of in-vitro resistance selection to ceftazidime/avibactam in this resistotype. To the best of our knowledge, this is the first-ever study, assessing multiple aspects of in-vitro resistance development in clinical isolates of OXA-181- and OXA-232-expressing *K. pneumoniae*. The investigations carried out in this study include genetic basis for the acquired resistance and its impact, if any, on in-vitro fitness as well as in-vivo infectivity/resistance. Further, to gain insights on the interplay between various β-lactams and β-lactamase inhibitors in overcoming ceftazidime/avibactam resistance in OXA-48-like pathogens, MICs of three cefepime based combinations under clinical development, two exploratory cefepime/BLI combinations, as well as various exploratory combinations of ceftazidime with BLI or β-lactam enhancer, were determined. The BLIs used in the study belonged to three different chemotypes: diazabicyclooctane, boronate and β-lactam.

One of the findings of this study was that the selected resistance to ceftazidime/avibactam, emanating from mutations in genes encoding AmpC, efflux, porins and stress response pathways, remained stable despite multiple consecutive drug-free passages, suggesting potential for ease of survival and dissemination of such high-risk resistant clones. Previous studies showed that mutations in these stress response pathway proteins could modulate efflux/impermeability [26,27,28,29], and hence, changes in porin (down-regulation) and efflux (up-regulation) protein expressions stand out as the possible mechanisms for ceftazidime/avibactam resistance. In a similar serial transfer study reported by Livermore et al. in 2015, stably raised MICs of ceftazidime/avibactam against KPC-expressing Enterobacterales were linked with mutations in Ω loop of KPC [20]. However, in contrast, our study did not show mutations in blaOXA-48-like; rather, mutations were mostly associated with efflux and porin encoding genes (barring a point mutation converting blaOXA-181 to catalytically comparable blaOXA-232). In another study by Livermore et al., exposure of ceftazidime/avibactam to de-repressed AmpC harboring *E. coli* and *K. pneumoniae*, led to the selection of mutants with stably elevated MICs (4–64 mg/L) and genomic studies showed several modifications in AmpC [22]. In the same study, ceftazidime/avibactam-resistant mutants of ESBL producers (ceftazidime/avibactam MICs 0.5–16 mg/L) mostly had changes affecting permeability, efflux or β-lactamase quantity; only one had an altered β-lactamase. However, in another study, polymorphism in blaCTX-M-14 has been implicated in clinical resistance to ceftazidime/avibactam which was due to augmented ceftazidimase activity of the mutated enzyme [30]. Thus, overall, it seems that established resistance imparting strategies involving mutation in β-lactamase genes as well as efflux and porins are the drivers of ceftazidime/avibactam resistance in OXA-48-like expressing strains.

In context to our study, it is pertinent to discuss a sole recent publication from Fröhlich et al. studying the propensity of resistance development in *E. coli* constructs expressing OXA-48 [31]. Similar to our study, Fröhlich and colleagues also described the ease of resistance selection upon exposure to ceftazidime/avibactam. However, unlike our study, mutations were reported in blaOXA-48 which resulted in increased hydrolysis of ceftazidime and reduced inhibitory activity of avibactam. This divergent observation could be linked to several factors such as differences in method of resistance selection (agar method vs. broth) and more importantly, the genus harboring blaOXA-48-like enzymes (*E. coli* vs. *K. pneumoniae*). Moreover, the two studies differed in terms of gene/s depicting the mutations (blaOXA-48 vs. blaCMY, efflux pumps and outer membrane porins). More pertinently, the present study employed clinical isolates harboring OXA-48-like enzymes along with impermeability and AmpC/ESBL, and therefore, possibly is more relevant to the real-world scenario.

From the evolutionary viewpoint, the success of Gram-negative pathogens in countering β-lactams through unleashing mutant β-lactamase gene could be judged by the fact that 2800 discrete β-lactamases [32] with wide range of substrate specificities have already been reported. Likewise, mutations in genes encoding/regulating porins and efflux pumps have also been successfully deployed with minimal fitness cost trade-off. This observation was substantiated in our study, as ceftazidime/avibactam-resistant mutants did not show any growth defects in-vitro as well as in-vivo Moreover, these mutants showed uncompromised ability to cause pulmonary infection in neutropenic mice and as anticipated, required elevated ceftazidime/avibactam doses for the protection of infected mice. Thus, ceftazidime/avibactam resistance observed in-vitro also manifested in-vivo, pointing towards the significance of our findings.

Since therapeutic options for infections caused by OXA-48-like phenotypes are limited, we investigated the potential of several combinations of chemically diverse BLIs/β-lactam enhancer zidebactam with cefepime or ceftazidime. In general, ceftazidime/avibactam-resistant mutants showed cross resistance to all the cephalosporin based BL/BLI combinations employed in this study, suggesting that despite deployment of structurally and mechanistically diverse BLIs, the BL/BLI mechanism of action has an intrinsic limitation in overcoming mutational resistance impacting other members of its own class (β-lactam antibiotics). Yet, as compared to ceftazidime, cefepime combinations tended to show lower MICs irrespective of partner BLI used in the combination. For instance, among them, cefepime/avibactam was the most potent combination, a benefit accruing from mechanistically advantageous features of cefepime as a partner β-lactam antibiotic. It has been reported that cefepime is bestowed with a β-lactamase stability, multiple PBP binding action and optimal permeation in Gram-negatives [33]. Other BLIs such as relebactam, enmetazobactam, taniborbactam and vaborbactam yielded inconsistent MICs in combination with either cefepime or ceftazidime. Previously, against mutant KPC-expressing Enterobacterales, a combination of avibactam with ceftaroline has been shown to possess better in-vitro activity as compared to its combination with ceftazidime [20]. This was ascribed to the ability of ceftaroline to better withstand the KPC-mediated hydrolysis. The role of partner antibiotic was also evident in our study, which showed that when imipenem was paired with avibactam, the majority of mutants turned susceptible to the combination. Thus, avibactam’s well-established feature of inhibition of OXA-48-like β-lactamases could be better harnessed in combination with imipenem, as compared to cephalosporins.

Finally, it was only the β-lactam enhancer, zidebactam, which provided a narrow range of low MICs regardless of the partner cephalosporin used. This is an outcome of the reported unconventional mechanism of action of zidebactam that operates through PBP2 binding and in combination with PBP3 binding β-lactams, which overcomes multiple mechanisms of resistance that adversely impact range of BL/BLI combinations [34,35,36,37]. For instance, unlike newer BLIs, the MICs of zidebactam in combination with relatively less potent partner ceftazidime were still below 8 mg/L against all the mutants.

In conclusion, the present study reveals the risk of resistance development with ceftazidime/avibactam against blaOXA-48-like-expressing *K. pneumoniae*, thereby mandating careful monitoring. Moreover, the study showed that acquisition of ceftazidime/avibactam resistance in *K. pneumoniae* is mediated through diverse mutational events in genes encoding efflux/impermeability without concomitant fitness cost suggesting possibility of successful survival and dissemination of such mutants. Sub-optimal activity of newer BLI based combinations points towards the multiplicity of resistance mechanisms in ceftazidime/avibactam selected mutants and limitations of BL/BLI approach in overcoming such resistance. The study also revealed the ability of β-lactam enhancer zidebactam in overcoming resistance to diverse BL/BLIs. This observation reinforces the need to continue efforts towards identifying novel Gram-negative therapies with unconventional mechanism of action.

## 4. Materials and Methods

### 4.1. Media, Antibiotics and Strains

Mueller–Hinton agar (MHA), cation-adjusted Mueller–Hinton broth (CAMHB) and agar were procured from Difco (Becton Dickinson, Franklin Lakes, NJ, USA. Tryptone soy broth was from Hi-media, India and was used for Tryptic soy agar (TSA) preparation. Ceftazidime, cefepime, imipenem and meropenem were acquired from commercial manufacturers. Various β-lactamase inhibitors and β-lactam enhancer (zidebactam) used in this study were kind gifts from Wockhardt Research Center, India. Ten *K. pneumoniae* included in the study were collected from Indian tertiary care hospitals during 2018, and were genomically characterized for *bla*OXA-181/*bla*OXA-232, ESBLs and class C β-lactamases, based on WGS. The species-level identification for all the parent strains and their respective mutants was undertaken by using VITEK 2.

### 4.2. Antimicrobial Susceptibility Testing

MIC of ceftazidime/avibactam and various comparator antibiotics (range used: 0.06 to 128 mg/L) was determined following the reference broth microdilution method as described in the Clinical & Laboratory Standards Institute [38,39]. Reference strains such as *K. pneumoniae* ATCC 700603, *K. pneum*oniae ATCC 1705 and *P. aeruginosa* ATCC 27853 were included during each MIC testing. For the sake of comparison purposes, parent and mutant strains were adjudged for their susceptibility to cefepime-based combinations employing susceptibility dose-dependent (SDD) breakpoint of cefepime and for ceftazidime-based combinations, susceptibility breakpoint of ceftazidime-avibactam was employed (both ≤8 mg/L).

### 4.3. Serial Transfer Studies

The study was initiated by determining macro-broth (2 mL volume of CAMHB) MICs for ceftazidime/avibactam against ten OXA-181- or OXA-232-expressing *K. pneumoniae* isolates. This was followed by sequential (every 24 h) transfer of inoculum (2–5 × 10^5^ CFU/mL) from 0.25× MIC for the next round of MIC determination [40]. Such sequential transfer was undertaken for 15 days in duplicate. After completion of serial passage cycles, cultures recovered after 5, 10 and 15 day passage were subjected to one drug free passage and preserved at −80 °C.

In order to assess the stability of resistance acquired after sequential ceftazidime/avibactam exposures, mutants were subjected to 15 drug-free passages on TSA followed by determination of MICs of the selecting agents.

### 4.4. In-Vitro Growth Assessments of Mutants

Comparative growth profiles of terminal mutants of ceftazidime/avibactam with their respective parent strains were determined in M9 minimal media (prepared by mixing NH_4_Cl, 5 g/L; KH_2_PO_4_, 15 g/L; Na_2_HPO_4_·7H_2_O, 64 g/L; NaCl, 2.5 g/L supplemented with glucose, 0.4%; CaCl_2_, 0.1 mM; MgSO_4_, 2 mM & pH 7.2) and CAMHB. Overnight grown bacterial cultures were diluted (1:10) in MHB and grown at 37 °C at 180 rpm till the cultures reach mid-log phase. The cultures were then diluted in M9 minimal medium and CAMHB (approximately 10^5^ CFU/mL) and grown under ambient conditions. The growth was monitored by viable cell enumeration on TSA at various time intervals till 12 h.

### 4.5. Molecular Characterization of Resistance Mechanisms

In order to identify the mutational changes in genes encoding β-lactam impacting enzymatic as well as non-enzymatic resistance, whole genome sequencing (WGS) was performed for terminal mutants and their respective parent strains. In short, genomic DNA (gDNA) was extracted by the alkaline lysis method and quantified by spectrophotometry (Schimadzu). gDNA libraries were prepared by using NEBNext Ultra DNA Library Preparation Kit. Such gDNA libraries were enriched on Agilent DNA HS chip and sequenced by Illumina MiSeq platform (San Diego, CA, USA). Genome sequences were assembled by SOAPdenovo2 v2.0.4 software. The genome sequence of each mutant was compared with the respective parent to identify single nucleotide polymorphisms (SNPs) by genome analysis tool kit. The whole genome sequencing data of all the parent and their respective mutant (except mutant of *K. pneumoniae* S471 under submission) isolates is deposited at DDBJ/ENA/GenBank under the accession JAHUX(R to Z)000000000 and JAHUY(A to J)000000000.

### 4.6. Infectivity and Resistance Studies in Murine Peritonitis Model Employing Ceftazidime/Avibactam-Resistant Mutants

In order to assess the impact of in-vitro resistance to ceftazidime/avibactam in blaOXA-48-like-harbouring *K. pneumoniae* on in-vivo infectivity and expression of resistance, murine peritonitis model was employed. Healthy male or female Swiss albino mice (18–20 g) were utilized for the in-vivo studies. In-vivo study protocols were reviewed and approved by Wockhardt’s animal ethics committee, registered under Committee for the Purpose of Control and Supervision of Experiments on Animals (CPCSEA), Government of India.

Male/female Swiss albino mice were infected through intra-peritoneal injection with bacterial inoculum (1–3 × 10^6^ CFU/mL) in 0.5 mL volume prepared in 5% hog gastric mucin (Sigma-Aldrich, St. Louis, MO, USA). Both parent and its respective ceftazidime/avibactam selected mutants of two *K. pneumoniae* strains (CMC307 and CMC387) were employed in this study. Two hours post-infection, animals (*n* = 6 per group) were dosed subcutaneously with ceftazidime in combination with avibactam at 4:1 ratio at different doses (dose rage of ceftazidime-3.12, 6.25, 12.5, 25, 50, 100 mg/kg; avibactam-0.78, 1.56, 3.12, 6.25, 12.5, 25 mg/kg) (1 dose per group) in 0.2 mL saline. Similarly, meropenem at 100 mg/kg dose (*n* = 6 animals) was administered. All the antibiotics were administered twice with 5 h interval for one day. Groups of animals (*n* = 6 per group) infected with parent and respective mutants without drug treatment were treated as placebo control to help assess the effect of in-vitro resistance on in-vivo infectivity.

In this model of infection, post-infection the mice from untreated group developed septicemia and became moribund within 24 h of infection unless they received adequate dose of antibiotic therapy. The efficacy of the antibiotic used in this study was measured using survival as the endpoint, with observation continued for 7 days post one day of antibiotic treatment.

The 50% effective dose (ED50) for the ceftazidime/avibactam treatment arm, were reported in terms of unit dose of ceftazidime. For instance, a 4:1 ceftazidime/avibactam combination ED50 of 20 mg/kg represents 20 mg/kg ceftazidime + 5 mg/kg avibactam. The ED50 values were calculated by log-probit analysis [41].

## Figures and Tables

**Figure 1 antibiotics-10-01318-f001:**
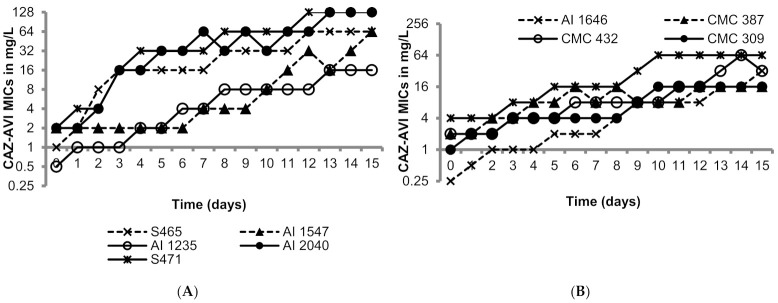
Serial transfer studies of ceftazidime/avibactam against ten OXA-48-like-expressing *K. pneumoniae.* (**A**) OXA-48 producing *K. pneumoniae* isolates had ceftazidime-avibactam MICs ranging from 0.5 to 1 mg/L. (**B**) OXA-48 producing *K. pneumoniae* isolates had ceftazidime-avibactam MICs ranging from of 0.5 to 4 mg/L.

**Figure 2 antibiotics-10-01318-f002:**
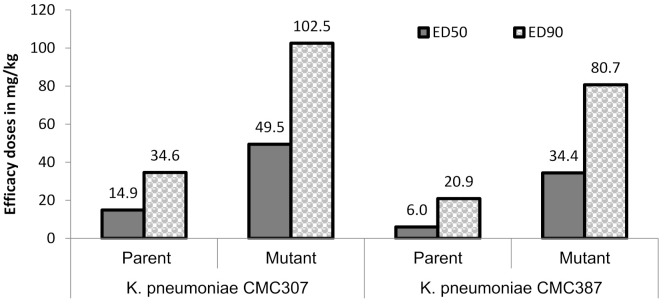
Efficacy of ceftazidime/avibactam in murine systemic infection model. Ceftazidime was administered (b.i.d) in multiple doses of 3.12, 6.25, 12.5, 25, 50, 100 mg/kg in combination with avibactam 4:1 dose of 0.78, 1.56, 3.12, 6.25, 12.5, 25 mg/kg, respectively. The lower limit-upper limit for ceftazidime ED_50_ and ED_90_ were 10.1–22.0 mg/kg (parent), 32.7–68.2 mg/kg (mutant) and 24.9–47.9 mg/kg (parent), 60–180.7 mg/kg (mutant), respectively, against *K. pneumoniae* CMC307. Similarly, against *K. pneumoniae* CMC387, they were 3.4–10.7 mg/kg, 24.3–48.8 mg/kg and 9.7–45.2 mg/kg, 48.4–134.7 mg/kg, respectively. All the animals in meropenem arm succumbed to death at the highest dose (100 mg/kg) tested.

**Table 1 antibiotics-10-01318-t001:** Characteristics of *K. pneumoniae* expressing OXA-48-like β-lactamases employed in serial transfer study.

Isolate ID	β-Lactamases	Ceftazidime-Avibactam MIC (mg/L)
S465	SHV-1, TEM-1, CTX-M-15, OXA-1, CMY-4, OXA-181	1
AI1547	SHV-1, TEM-1, CTX-M-15, DHA-1, CMY-4, OXA-232	1
AI1235	SHV-1, TEM-1, CTX-M-15, OXA-1, OXA-18, OXA-232	0.5
AI2040	SHV-1, TEM-1, CTX-M-15, OXA-232	1
S471	SHV-1, TEM-1, CTX-M-15, DHA-1, CMY-4, OXA-181	1
AI1646	SHV-1, TEM-1, CTX-M-15, OXA-232	0.5
CMC387	SHV-1, TEM-1, CTX-M-15, OXA-1, OXA-18, OXA-232	2
CMC432	SHV-1, TEM-1, CTX-M-15, OXA-1, OXA-232	2
CMC309	SHV-1, OXA-1, OXA-232	2
CMC307	SHV-1, TEM-1, CTX-M-15, OXA-1, OXA-18, OXA-10, OXA-181	4

**Table 2 antibiotics-10-01318-t002:** List of mutations identified in terminal mutants obtained from ceftazidime/avibactam serial exposure.

Isolate Name	Ceftazidime-Avibactam MIC (mg/L)	Mutated Gene	Gene Length (bp)	Mutation in DNA Sequence	Change in Protein	Probable Reason(s) of Ceftazidime/Avibactam Resistance
S465	1	*cmy-4*	1146	G_535_A	G179S	β-lactamase
AI1547	1	*kpnG* *mdtA* *mdtB*	117311103123	A_304_G A_719_T T_1751_G	K102E Q240L I584S	efflux
AI1235	0.5	*mrdA* *cpxA*	19021374	A_1061_C G_271_A	N354A E91K	efflux/impermeability
AI2040	1	*rpoE*	576	C_497_T	P166L	efflux/impermeability
S471	1	*envZ* *ompk36*	13561125	A_998_T T_1109_A	H333L L370Q	impermeability
AI1646	0.5	*acrD*	3114	G_862_A	G288S	efflux
CMC387	2	*acrB* *cpxA*	31471374	T_1897_A G_1243_A	W633R G415S	efflux, efflux/impermeability
CMC432	2	*envZ*	1356	T_116_G	I39S	efflux/impermeability
CMC309	2	*acrB*	3147	T_416_G	V139G	efflux
CMC307	4	*acrB*	3147	A_403_C; T_1196_A	S135R V399E	efflux

Two blaOXA-181-harbouring isolates (*K. pneumoniae* AI1547 and *K. pneumoniae* S471) selected point mutations leading to conversion of blaOXA-181 to blaOXA-232. Previous reports established that ceftazidime/avibactam is stable to OXA-232, and therefore, this particular mutational event cannot be the reason for ceftazidime/avibactam resistance.

**Table 3 antibiotics-10-01318-t003:** Comparative activity of cefepime/ceftazidime when combined with various BLIs against ceftazidime-avibactam-resistant terminal mutants of *bla*_OXA-181_ and *bla*_OXA-232_ expressing *K. pneumoniae* recovered from serial transfer studies.

Isolate ID	Parent/ Mutant	CAZ	CAZ + AVI-4	FEP	FEP + AVI-4	FEP + REL-4	FEP + TAN-4	FEP + ZID 1:1	FEP + VAB-8	FEP + ENME-8	MEM	IPM	IPM + REL-4	IPM + AVI-4	ZID	CAZ + REL-4	CAZ + TAN-4	CAZ + ZID 1:1	CAZ + VAB-8	CAZ + ENME-8
S465	Parent	>128	1	64	0.5	2	4	2	8	8	32	8	8	0.12	4	4	1	2	128	>128
	Day15 Mutant	>128	32	128	0.5	1	2	4	16	8	8	8	8	0.25	4	128	128	8	>128	>128
AI 1547	Parent	>128	1	>128	1	4	1	1	16	8	32	4	2	0.25	>128	8	4	1	64	32
	Day15 Mutant	>128	128	128	1	4	2	1	16	16	32	8	8	0.25	>128	>128	>128	1	0.25	32
AI 1235	Parent	>128	0.5	>128	1	8	4	1	32	16	32	4	2	0.25	2	8	1	0.5	16	8
	Day15 Mutant	>128	8	>128	16	32	16	2	128	16	16	16	16	0.5	2	32	16	2	64	64
AI 2040	Parent	>128	1	>128	1	4	2	2	16	32	32	4	2	0.12	>128	4	2	2	8	8
	Day15 Mutant	>128	64	>128	16	64	64	8	128	>128	128	32	16	0.5	>128	128	128	8	>128	>128
S 471	Parent	>128	1	>128	1	4	4	1	8	4	32	8	4	0.5	>128	2	16	1	64	>128
	Day15 Mutant	>128	64	>128	16	>128	32	8	>128	>128	128	8	4	1	>128	>128	128	4	>128	>128
AI 1646	Parent	>128	0.5	>128	0.25	2	1	0.5	16	8	32	4	4	0.06	1	2	1	0.5	8	4
	Day15 Mutant	>128	16	>128	0.25	1	0.25	0.5	32	4	0.5	0.25	0.5	0.5	0.5	16	16	0.5	32	16
CMC 387	Parent	>128	2	>128	2	32	8	2	32	32	64	8	8	0.12	>128	8	8	1	16	8
	Day15 Mutant	>128	16	128	8	8	2	2	128	32	64	16	8	0.12	>128	32	8	2	64	16
CMC 432	Parent	>128	2	>128	2	8	8	1	32	32	32	8	4	0.5	>128	4	4	1	16	8
	Day15 Mutant	>128	16	128	8	16	8	1	32	16	32	4	2	1	>128	16	16	1	32	16
CMC 309	Parent	4	2	8	1	4	4	0.5	8	8	16	2	2	0.5	>128	2	0.5	0.25	4	4
	Day15 Mutant	32	16	64	2	4	4	0.5	16	8	8	2	1	0.5	>128	8	8	0.25	16	8
CMC 307	Parent	>128	4	>128	2	16	8	2	64	32	64	16	8	0.06	>128	16	8	2	64	64
	Day15 Mutant	>128	64	>128	8	32	8	2	>128	16	64	8	8	8	>128	64	32	2	>128	32

CAZ: ceftazidime, AVI: avibactam (fixed 4 mg/L), FEP: cefepime, TAN: taniborbactam (fixed 4 mg/L). REL: relebactam (fixed 4 mg/L), VAB: vaborbactam (fixed 8 mg/L), ENME: enmetazobactam (fixed 8 mg/L), IPM: imipenem, MEM: meropenem, FEP-ZID (cefepime-zidebactam 1:1 fixed ratio), CAZ-ZID (ceftazidime-zidebactam 1:1 fixed ratio).

## Data Availability

The study did not report any data.

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
