# Peer review of "In-Vitro Selection of Ceftazidime/Avibactam Resistance in OXA-48-Like-Expressing Klebsiella pneumoniae: In-Vitro and In-Vivo Fitness, Genetic Basis and Activities of β-Lactam Plus Novel β-Lactamase Inhibitor or β-Lactam Enhancer Combinations"

_antibiotics, 2021, doi:10.3390/antibiotics10111318_

Round 1
Reviewer 1 Report
The manuscript is well written. Design and results are clearly exposed. Discussion section is right and provides important data. However, authors use the results section to discuss some points and support their results with references. The correct section to do this is the discussion.
Other minor comments:
The authors do not describe the acronyms the first time they name them (they do so in the materials section at the end).
Add the mutation appearing in OXA-181 (change from OXA-181 to OXA-232) in Table 2.
Author Response
Authors’ responses to reviewers’ comments
Manuscript ID: antibiotics-1401955
We thank the editor and reviewers for their review and helpful comments on our manuscript.
Below, please find our point-by-point responses to reviewers’ comments.
Reviewer 1
Main comment
Authors use the results section to discuss some points and support their results with references. The correct section to do this is the discussion.
Authors’ response
While we are respectful to the reviewer’s comment, we think, for the reader’s better understanding on the functional role of the mutations detected in our analyses, citing necessary previous reports at few instances was necessary. However, based on the reviewer’s suggestion, text pertaining to previously described role of stress response in modulating the efflux and porin has been shifted to the discussion section.
Other comments
The authors do not describe the acronyms the first time they name them (they do so in the materials section at the end).
Authors’ response
We regret for this oversight and as suggested appropriate descriptions for acronyms are now provided when they appear first in the text.
Add the mutation appearing in OXA-181 (change from OXA-181 to OXA-232) in Table 2.
Authors’ response
In the table, we summarized only those mutations which potentially contribute to the rise in the MICs of ceftazidime/avibactam. With respect to the change from OXA-181 to OXA-232 found in terminal mutants of two strains (AI1547 and S471), previous reports established that ceftazidime/avibactam is stable to OXA-232 and therefore this particular mutational event is highly unlikely to be the reason for ceftazidime/avibactam resistance. However, to address the reviewer’s comment, we added this information in the foot note of Table 2.
Reviewer 2 Report
In Vitro Selection of Ceftazidime/Avibactam Resistance in 2 OXA-48-like-expressing Klebsiella pneumoniae: In Vitro and In Vivo Fitness, Genetic Basis and Activities of β-lactam plus Novel β-lactamase Inhibitor or β-lactam Enhancer Combinations
Technical Comments to the Author
The manuscript is written in clear and understandable English, the results are more very fluent. Furthermore, the connection among the results and the figures/tables as well as final message is very strong.
Remarks to the Author
I suggest minor comments.
Minor comments
1) Provide more information on the methods used to identify the bacterial strains under study.
2) Strains of Klebsiella pneumoniae were genomically characterized for blaOXA-181/blaOXA-232, extended spectrum β-lactamases (ESBLs) and class C β-lactamases. Attach the method used.
3) What concentration ranges have been tested for ceftazidime-avibactam and various comparator antibiotics? (Material and methods section, paragraph 4.2).
4) Assign references to the serial transfer method used (Material and methods section, paragraph 4.3)
5) Clearly describe in material and methods section (paragraph 4.6) the mice groups used in this study. How many groups are used? how many animals does each group include? what is the role of each group?
6) Report in vitro and in vivo in italics (lines 23, 215).
7) Double space is present in line 43 of the discussion section.
Author Response
Authors’ responses to reviewers’ comments
Manuscript ID: antibiotics-1401955
We thank the editor and reviewers for their review and helpful comments on our manuscript.
Below, please find our point-by-point responses to reviewers’ comments.
Reviewer 2
No major comments
Minor comments
Provide more information on the methods used to identify the bacterial strains under study
Authors’ response
Bacterial strains were identified by VITEK 2 and same has been mentioned in the revised manuscript.
Strains of Klebsiella pneumoniae were genomically characterized for blaOXA-181/blaOXA-232, extended spectrum β-lactamases (ESBLs) and class C β-lactamases. Attach the method used.
Authors’ response
All the β-lactamases genes were identified based on whole genome sequencing and now mentioned in the ‘Materials and Methods’ section.
What concentration ranges have been tested for ceftazidime-avibactam and various comparator antibiotics? (Material and methods section, paragraph 4.2).
Authors’ response
A concentration range of 0.06 to 128 mg/L, in doubling dilutions was tested. This information has been added in the revised manuscript.
Assign references to the serial transfer method used (Material and methods section, paragraph 4.3)
Authors’ response
As suggested, we have included a reference for the serial transfer method.
Clearly describe in material and methods section (paragraph 4.6) the mice groups used in this study. How many groups are used? how many animals does each group include? what is the role of each group?
Authors’ response
We used n = 6 mice per group. There was one untreated group, and multiple ceftazidime/avibactam treated groups (doses from 3.12/0.78 mg/kg to 100/25 mg/kg) and one meropenem treated group (dose of 100 mg/kg). We have revised the relevant text for better clarity.
Report in vitro and in vivo in italics (lines 23, 215)
Authors’ response
The terms in vitro and in vivo are in italics in the revised manuscript.
Double space is present in line 43 of the discussion section.
Authors’ response
The space error has been corrected in the revised manuscript.
Reviewer 3 Report
The article "In-Vitro Selection of Ceftazidime/Avibactam Resistance in OXA-48-like-expressing Klebsiella pneumoniae: In Vitro and In Vivo Fitness, Genetic Basis and Activities of β-lactam plus Novel β-lactamase Inhibitor or β-lactam Enhancer Combinations" shows the effect of OXA-48-like carbapenemase-expressing Klebsiella pneumoniae on development of resistance in time dependent manner along with its in vivo efficacy studies. The work seems to be and extension of Livermore et al 2015 and Fröhlich et al., 2019 but is of significant importance in the field of antibiotic resistance as well as in-vivo efficiency of antibiotics with respect to development of resistant mutants.
Abstract:
The work explains the proper objective and supports it with conclusive results.
Introduction:
The authors have raised a proper issue associated with the use of combinational use of antibiotics and the problems associated with it in development of resistance. The introduction is well supported with recent and updated review of literature to support the rational of the performed work. The only limitation seems to be associated with the possible background of similar previous studies carried out in this areas like those of Livermore et al 2015 and Fröhlich et al., 2019. Also, considering the previously reported reviews from Bush and Bradford, 2016 and Tooke, 2019, this works seems to less explainatory as far as introduction is considered. I would suggest the authors to improvise the introduction section to make it more complete. Introduction lacks the depth of review which it requires to further support its rational!
Results:
The serial transfer study clearly shows the dose dependent nature of the 10 different OXA-48-like-expressing K. pneumoniae isolates and the whole genome sequencing associated mutation of genes are well supported with previous work but it seems to have an abrupt ending as far as line 131-134 is concerned. The author must provide some explanation with respect to the effect of these mutations with respect to previously reported studies or a hypothesis to make it more conclusive.
The invivo studies are showing significant dose dependent response as far as development of resistance in mutants is concerned and validates the findings. But how would the authors like to defend the findings with mutants as far as line 157 and 158 are concerned......... "All the animals in meropenem arm succumbed to death at the highest 157 dose (100 mg/kg) tested"
The Susceptibility profile associated with combination of antibiotics and with β-lactamase inhibitors/β-lactam enhancer explains their interdependence and for some isolate its significant.
Discussion:
The serious issue of development of antibiotic resistance supported with in vitro and in vivo results is discussed well in this reported study and the the limitation of antibiotic combinations for treatments are evident. Considering the scope of this journal as well as the quality of the reported findings, the article can be considered for publication following suggested minor changes as well as other reviewers expert comments.
Author Response
Authors’ responses to reviewers’ comments
Manuscript ID: antibiotics-1401955
We thank the editor and reviewers for their review and helpful comments on our manuscript.
Below, please find our point-by-point responses to reviewers’ comments.
Reviewer 3
Major comments
Introduction: The only limitation seems to be associated with the possible background of similar previous studies carried out in this areas like those of Livermore et al 2015 and Fröhlich et al., 2019. Also, considering the previously reported reviews from Bush and Bradford, 2016 and Tooke, 2019, this works seems to less explainatory as far as introduction is considered. I would suggest the authors to improvise the introduction section to make it more complete. Introduction lacks the depth of review which it requires to further support its rational!
Authors’ response
The reviewers concern was addressed by providing a revamped introduction section in the revised submission. Specifically, we have touched upon previously reported resistance development studies which were majorly undertaken employing KPC expressing K. pneumoniae and have highlighted that unlike a singular study employing OXA48-like constructs, our study is based on ‘real-world’ K. pneumoniae isolates.
Results: The serial transfer study clearly shows the dose dependent nature of the 10 different OXA-48-like-expressing K. pneumoniae isolates and the whole genome sequencing associated mutation of genes are well supported with previous work but it seems to have an abrupt ending as far as line 131-134 is concerned. The author must provide some explanation with respect to the effect of these mutations with respect to previously reported studies or a hypothesis to make it more conclusive.
Authors’ response
As suggested we completed the text by explaining the potential reason for lack of change in ceftazidime/avibactam MICs due to PBP2 mutation in K. pneumoniae. We also highlighted that depending on the base line MICs of ceftazidime/avibactam, a variable no of passages were required to raise the MICs beyond the susceptibility breakpoint of 8 mg/L.
The in vivo studies are showing significant dose dependent response as far as development of resistance in mutants is concerned and validates the findings. But how would the authors like to defend the findings with mutants as far as line 157 and 158 are concerned......... "All the animals in meropenem arm succumbed to death at the highest 157 dose (100 mg/kg) tested"
Authors’ response
Both the parent strains (CMC307 and CMC 387) studied in vivo harboured carbapenem-resistance and their mutants also retained the carbapenem-resistance phenotype. Therefore, infections caused by these meropenem-resistant strains did not respond to meropenem treatment, thus resulting in uncontrolled lung infection leading to mortality.